# Task-agnostic Exploration in Reinforcement Learning

**Xuezhou Zhang**
UW-Madison
xzhang784@wisc.edu

**Yuzhe Ma**
UW-Madison
ma234@wisc.edu

**Adish Singla**
MPI-SWS
adishs@mpi-sws.org

## Abstract

Efficient exploration is one of the main challenges in reinforcement learning (RL). Most existing sample-efficient algorithms assume the existence of a single reward function during exploration. In many practical scenarios, however, there is not a single underlying reward function to guide the exploration, for instance, when an agent needs to learn many skills simultaneously, or multiple conflicting objectives need to be balanced. To address these challenges, we propose the *task-agnostic RL* framework: In the exploration phase, the agent first collects trajectories by exploring the MDP without the guidance of a reward function. After exploration, it aims at finding near-optimal policies for $N$ tasks, given the collected trajectories augmented with *sampled rewards* for each task. We present an efficient task-agnostic RL algorithm, UCBZERO, that finds $\varepsilon$-optimal policies for $N$ arbitrary tasks after at most $\tilde{O}(\log(N)H^5 SA/\varepsilon^2)$ exploration episodes, where $H$ is the episode length, $S$ is the state space size, and $A$ is the action space size. We also provide an $\Omega(\log(N)H^2 SA/\varepsilon^2)$ lower bound, showing that the $\log$ dependency on $N$ is unavoidable. Furthermore, we provide an $N$-independent sample complexity bound of UCBZERO in the recently proposed *reward-free* setting, a statistically easier setting where the ground truth reward functions are known.

## 1 Introduction

Efficient exploration is widely regarded as one of the main challenges in reinforcement learning (RL). Recent works in theoretical RL have provided near-optimal algorithms in both model-based [Jaksch et al., 2010, Azar et al., 2017, Zanette and Brunskill, 2019] and model-free [Strehl et al., 2006, Jin et al., 2018, Zhang et al., 2020] paradigms, that are able to learn a near-optimal policy with a sample complexity that matches the information-theoretical lower bound. However, these algorithms are designed to solve a single pre-defined task and rely on the assumption that a well-defined reward function is available during exploration. In such settings, policy optimization can be performed simultaneously with exploration and results in an effective exploration-exploitation trade-off.

In many real-world applications, however, a pre-specified reward function is not available during exploration. For instance, in recommendation systems, reward often consists of multiple conflicting objectives, and the balance between them is tuned via continuous trial and error to encourage the desired behavior [Zheng et al., 2018]; In hierarchical and multi-task RL [Dietterich, 2000, Tessler et al., 2017, Oh et al., 2017], the agent aims at simultaneously learning a set of skills; In the robotic navigation problem [Rimon and Koditschek, 1992, Kretzschmar et al., 2016], the agent needs to navigate to not only one goal location, but a set of locations in the environment. Motivated by these real-world challenges, we ask: *Is it possible to efficiently explore the environment and simultaneously learn a set of potentially conflicting tasks?*

To answer this question, we present the *task-agnostic RL* paradigm: During the exploration phase, the agent collects trajectories from an MDP *without* the guidance of pre-specified reward function. Next, in the policy-optimization phase, it aims at finding near-optimal policies for $N$ tasks, given the

| Setting | No. of tasks | Upper bound | Lower bound |
|---|---|---|---|
| **Task-specific RL** | $N = 1$ | $\tilde{O}\left(\frac{H^2 SA}{\varepsilon^2}\right)$ [ORLC] | $\Omega\left(\frac{H^2 SA}{\varepsilon^2}\right)$ |
| **Reward-free RL** | $N$ dep. | $\tilde{O}\left(\frac{\log(N) H^5 SA}{\varepsilon^2}\right)$ [UCBZERO] | - |
| | $N$ indep. | $\tilde{O}\left(\frac{H^5 S^2 A}{\varepsilon^2}\right)$ [RFE] | $\Omega\left(\frac{H^2 S^2 A}{\varepsilon^2}\right)$ |
| **Task-agnostic RL** | $N$ dep. | $\tilde{O}\left(\frac{\log(N) H^5 SA}{\varepsilon^2}\right)$ [UCBZERO] | $\Omega(\frac{\log(N) H^2 SA}{\varepsilon^2})$ |
| | $N$ indep. | $\infty$ | $\infty$ |

Table 1: Comparison of sample complexity results in three RL settings. Shaded cells represent our results.

collected trajectories augmented with the *instantiated rewards* sampled from the unknown reward function for each task. We emphasize that our framework covers applications like recommendation systems, where stochastic objectives (click rate, dwell time, etc.) are observed with the transitions, but the ground-truth reward functions are not known. There is a common belief in task-specific RL that estimating the reward function is not a statistical bottleneck. We will show that, in the more challenging task-agnostic setting, the need to estimate rewards makes the problem strictly harder.

To address the task-agnostic RL problem, we present an efficient algorithm, UCBZERO, that can explore the MDP without the guidance of a pre-specified reward function. The algorithm design adopts the *Optimism Under Uncertainty* principle to perform exploration and uses a conceptually simpler Azuma-Hoeffding-type reward bonus, instead of a Bernstein-Friedman-type reward bonus that typically achieves better bounds in the task-specific RL framework. The advantage of an Hoeffding-type bonus is that it only depends on the number of visitations to the current state-action pair, but not on the reward value. This is a key property that allows it to be used in the task-agnostic setting. The UCBZERO algorithm is defined in Alg. 1. Despite its simplicity, UCBZero provably finds $\varepsilon$-optimal policies for $N$ arbitrary tasks simultaneously after at most $\tilde{O}(\log(N) H^5 SA/\varepsilon^2)$ exploration episodes. To complement our algorithmic results, we establish a near-matching $\Omega(\log(N) H^2 SA/\varepsilon^2)$ lower bound, demonstrating the near-optimality of UCBZERO as well as the necessity of the dependency on $N$, which is unique to the task-agnostic setting.

**Our Contributions:**

1. We propose a novel *task-agnostic RL* framework and present an algorithm, UCBZERO, that finds near-optimal polices to all $N$ tasks with small sample complexity. In addition, we provide a near-matching lower bound, demonstrating the near-optimality of our algorithm.
2. We investigate interesting properties of the UCBZERO algorithm that shine a light on its success. In particular, we show that (i) UCBZERO is able to **visit all state-action pairs** sufficiently often, and (ii) the transitions collected by UCBZERO allows one to obtain **accurate estimate of the dynamics model**.
3. Lastly, we provide an $N$-independent sample complexity bound of UCBZERO in a statistically easier setting studied in a contemporary work [Jin et al., 2020], where all ground truth reward functions are known. We provide a unified view of the two settings and present detailed discussion contrasting the statistical challenges between the two. Table 1 summarizes our key theoretical results.

## 2 Related Work

**Sample-efficient algorithms for the task-specific RL.** There has been a long line of work that design provably efficient algorithms for the task-specific RL setting. In the model-based setting, UCRL2 [Jaksch et al., 2010] and PSRL [Agrawal and Jia, 2017] estimates the transition dynamics of the MDP with upper-confidence bounds (UCB) added to encourage exploration. UCBVI [Azar et al., 2017] and later ORLC [Dann et al., 2018] improves the sample complexity bound to match with the information-theoretic lower bound $O(H^2 SA/\varepsilon^2)$ [Dann and Brunskill, 2015]. In the model-free setting, delayed Q-learning [Strehl et al., 2006] is the first sample efficient algorithm. UCB-H, UCB-B [Jin et al., 2018] and later UCB-Advantage [Zhang et al., 2020] improve upon previous results and matches the information-theoretical lower-bound up to log factors and lower-order terms. Our algorithm UCBZERO can be viewed as a zero-reward version of the UCB-H algorithm

and is shown to achieve the same sample complexity as UCB-H when there is a single task, despite not having access to reward information during exploration. Our results suggest that a zero-reward version of other sample-efficient algorithms, such as UCB-B and UCB-Advantage, can potentially be adapted to the task-agnostic setting to achieve tighter bounds.

**Empirical study of task-agnostic RL.** In the Deep Reinforcement Learning (DRL) community, there has been an increasing interest in designing algorithms that can learn without the supervision of a well-designed reward function. Part of the motivation comes from the ability of humans and other mammals to learn a variety of skills without explicit external rewards. Task-agnostic RL is studied in a variety of settings, including robotic control [Riedmiller et al., 2018, Finn and Levine, 2017], self-supervised model learning [Xie et al., 2018, Xie et al., 2019], general value function approximation (GVFA) [Schaul et al., 2015], multi-task meta-learning [Sæmundsson et al., 2018], etc. Even when one only cares about solving a single task, auxiliary navigation tasks have been shown to help solving problems with sparse rewards, in algorithms such as Hindsight Experience Replay (HER) [Andrychowicz et al., 2017] and Go-Explore [Ecoffet et al., 2019]. Our work provides a theoretical foundation for these empirically successful algorithms.

**Closely related contemporary work.** A closely related contemporary work proposes the *reward-free RL* setting [Jin et al., 2020]. Their setting differs from ours mainly in that they assume the availability of true reward functions during the policy optimization phase, whereas we only assume the availability of sampled rewards on the collected transitions. In the task-specific RL setting, having to estimate reward is typically not considered a statistical bottleneck, since estimating the reward function boils down to estimating an additional $SA$ parameters, which is negligible compared to the estimation of transition model of size $S^2 A$. However, when there are $N$ tasks, the parameters of reward functions now have a total size of $SAN$. When $N$ is large, this estimation error becomes non-negligible. Therefore, our setting can be statistically more challenging than the *reward-free* ssetting. We provide a thorough comparison between the two frameworks in Section 6.

## 3   Preliminaries

In this paper, we consider the setting of a tabular episodic Markov decision process, $\text{MDP}(\mathcal{S}, \mathcal{A}, H, P, r)$, where $\mathcal{S}$ is the state space of size $S$, $\mathcal{A}$ is the action space of size $A$, $H$ is the number of steps in each episode, $P$ is the time-dependent transition matrix so that $P_h(\cdot|s, a)$ gives the distribution over the next state if action $a$ is taken from state $s$ at step $h \in [H]$, and $r_h(\cdot|s, a)$ is a stochastic reward function at step $h$ whose support is bounded in $[0, 1]$. Without loss of generality, we assume that the MDP has a fixed starting state $s_1$. The general case reduces to this case by adding an additional time step at the beginning of each episode.

A policy $\pi$ is a collection of $H$ functions $\{\pi_h : \mathcal{S} \to \Delta_{\mathcal{A}}\}_{h \in H}$, where $\Delta_{\mathcal{A}}$ is the probability simplex over $\mathcal{A}$. We denote $V_h^{\pi} : \mathcal{S} \to \mathbb{R}$ as the value function at step $h$ under policy $\pi$, and $Q_h^{\pi} : \mathcal{S} \times \mathcal{A} \to \mathbb{R}$ as the action-value function at step $h$ under policy $\pi$, i.e.

$$V_h^{\pi}(s) = \mathbf{E}_{\pi} \left[ \sum_{h'=h}^{H} r_{h'}(s_{h'}, a_{h'}) | s_h = s \right], \quad Q_h^{\pi}(s, a) = \mathbf{E}_{\pi} \left[ \sum_{h'=h}^{H} r_{h'}(s_{h'}, a_{h'}) | s_h = s, a_h = a \right]$$

Since the total reward is finite, there exists an optimal policy $\pi^*$ that gives the maximum value $V_h^*(s) = \max_{\pi} V_h^{\pi}(s)$ for all $s \in S, h \in H$. Denoting $[\mathbb{P}_h V_{h+1}](s, a) := \mathbf{E}_{s' \sim P_h(\cdot|s,a)} [V_{h+1}(s')]$, we have the following Bellman's equation and Bellman's optimal equation:

$$V_h^{\pi}(s) = Q_h^{\pi}(s, \pi_h(s)), \qquad Q_h^{\pi}(s, a) = \mathbf{E}\left[r_h(s, a)\right] + \mathbb{P}_h V_{h+1}^{\pi}(s, a) \qquad (1)$$
$$V_h^*(s) = \max_a Q_h^*(s, a), \qquad Q_h^*(s, a) = \mathbf{E}\left[r_h(s, a)\right] + \mathbb{P}_h V_{h+1}^*(s, a) \qquad (2)$$

where we define $V_{H+1}^{\pi}(s) = 0$ for all $s \in S$. In this work, we evaluate algorithms based on the **sample complexity of exploration** framework [Kakade et al., 2003], where the goal is to find an $\varepsilon$-optimal policy $\pi$, i.e. $V_1^*(s_1) - V_1^{\pi}(s_1) \leq \varepsilon$, with probability at least $(1 - p)$. An algorithm is said to be **PAC-MDP** (Probably Approximately Correct in Markov Decision Processes) if for any $\varepsilon, p$, the sample complexity of finding an $\varepsilon$-optimal policy with probability at least $(1 - p)$ is $O(\text{poly}(H, S, A, 1/\varepsilon, 1/p))$.

**Task-agnostic RL.** In this paper, we study the setting of *task-agnostic RL* where learning is performed in two phases. In the *exploration phase*, the algorithm interacts with the enviornment for $K$ episodes

without the guidance of a reward, and collects a dataset of transitions $\mathcal{D} = \{(s_h^k, a_h^k)\}_{h \in [H], k \in [K]}$. In the *policy-optimization phase*, for each task $n \in [N]$, let $r_h^{(n)}(\cdot | s, a)$ be the unknown reward function for task $n$, and rewards are instantiated on the collected transitions, augmenting the dataset to be $\mathcal{D}^{(n)} = \{(s_h^k, a_h^k, r_h^k)\}_{h \in [H], k \in [K]}$, where $r_h^k \sim r_h^{(n)}(\cdot | s_h^k, a_h^k)$. The goal is to find $\varepsilon$-optimal policies for all $N$ tasks, and algorithms are evaluated based on the number of episodes $K$ needed to reliably achieve this goal. We remark that our setting generalizes and is more challenging than the standard task-specific RL setting and the *reward-free RL* setting in the contemporary work [Jin et al., 2020]; importantly, our algorithm and upper bounds apply to both those settings.

## 4 UCBZERO: Task-agnostic Exploration

---

**Algorithm 1** UCBZERO

---

**PARAMETERS:** No. of tasks $N$, $\iota = \log(SAHK/p)$, $b_t = c\sqrt{H^3(\log(N) + \iota)/t}$, $\alpha_t = \frac{H+1}{H+t}$.

**Exploration Phase:**

1: initialize $\overline{Q}_h(s, a) \leftarrow H$, $N_h(s, a) \leftarrow 0$ for all $(s, a, h) \in S \times A \times H$.
2: **for** episode $k = 1, ..., K$ **do**
3:     receive $s_1^k$.
4:     **for** step $h = 1, ..., H$ **do**
5:         Take action $a_h^k \leftarrow \arg\max_a \overline{Q}_h(s_h^k, a)$, and observe $s_{h+1}^k$.
6:         $t = N_h(s_h^k, a_h^k) \leftarrow N_h(s_h^k, a_h^k) + 1$. $\overline{V}_{h+1}(s_{h+1}^k) = \min(H, \max_a \overline{Q}_{h+1}(s_{h+1}^k, a))$.
7:         $\overline{Q}_h(s_h^k, a_h^k) \leftarrow (1 - \alpha_t)\overline{Q}_h(s_h^k, a_h^k) + \alpha_t[\overline{V}_{h+1}(s_{h+1}^k) + 2b_t]$.
8: **Return** $\mathcal{D} = \{(s_h^k, a_h^k)\}_{h \in [H], k \in [K]}$.

**Policy-Optimization Phase for Task n $\in$ [N]:**
**INPUTS:** task-specific reward-augmented transitions $\mathcal{D}^{(n)} = \{(s_h^k, a_h^k, r_h^k)\}_{h \in [H], k \in [K]}$.

1: initialize $\Pi = \emptyset$, $Q_h(s, a) \leftarrow H$, $N_h(s, a) \leftarrow 0$ for all $(s, a, h) \in S \times A \times H$.
2: **for** episode $k = 1, ..., K$ **do**
3:     $\Pi$.append($\pi_k$), where $\pi_k$ is the greedy policy w.r.t. the current $\{Q_h\}_{h \in [H]}$.
4:     **for** step $h = 1, ..., H$ **do**
5:         $t = N_h(s_h^k, a_h^k) \leftarrow N_h(s_h^k, a_h^k) + 1$, $V_{h+1}(s_{h+1}^k) = \min(H, \max_a Q_{h+1}(s_{h+1}^k, a))$.
6:         $Q_h(s_h^k, a_h^k) \leftarrow (1 - \alpha_t)Q_h(s_h^k, a_h^k) + \alpha_t[r_h^k + V_{h+1}(s_{h+1}^k) + b_t]$.
7: **Return** the (non-stationary) stochastic policy equivalent to uniformly sampling from $\Pi$.

---

We begin by presenting our algorithm, UCBZERO, as defined in Alg. 1. In the task-agnostic RL setting, the algorithm needs to handle both the *exploration phase* and the *policy-optimization phase*. In the exploration phase, UCBZERO maintains a pseudo-$Q$ table, denoted $\overline{Q}_h$, that estimates the cumulative UCB bonuses from the current step $h$. By acting greedily with respect to $\overline{Q}_h$ in step $h$, the algorithm will choose actions that can lead to under-explored states in future steps $h' > h$, i.e. states with large UCB bonus. In constrast to the original UCB-H algorithm which incorporates task-specific rewards and thus performs an exploration-exploitation trade-off, UCBZERO is a full-exploration algorithm and, therefore, is able to keep visiting all *reachable states* throughout the exploration phase.

In the policy-optimization phase, UCBZERO performs the same optimistic Q-learning update, only with smaller UCB bonuses than the ones in the exploration phase. For the sake of theoretical analysis, at the end of the policy-optimization phase, UCBZERO outputs a non-stationary stochastic policy equivalent to uniformly sampling from $\{\pi_k\}_{k \in [K]}$, the greedy policies w.r.t. the Q table at the beginning of each episode $k \in K$. We remark that both the exploration phase and policy-optimization phase of UCBZERO are **model-free**, enjoying smaller space complexity than model-based algorithms, including the RFE algorithm in the comtemporary work [Jin et al., 2020]. Next, we present our theoretical results upper-bounding the number of exploration episodes required for UCBZERO to find $\varepsilon$-optimal policies for $N$ tasks. Due to space constraints, we discuss key ideas in proving the theoretical results and defer the full proofs to the appendix.

### 4.1 Upper-bounding the Sample Complexity of UCBZERO

The exploration phase of UCBZERO is equivalent to a zero-reward version of the UCB-H algorithm. Our first result shows that, despite not knowing the reward functions during exploration, UCBZERO enjoys the same sample complexity as UCB-H in the task-agnostic setting, except for an additional $\log$ factor on the number of tasks $N$.

**Notation:** We denote by $Q_h^k$, $V_h^k$, $N_h^k$ respectively the $Q_h$, $V_h$, $N_h$ functions at the beginning of episode $k$, and similarly for $\overline{Q}_h^k$, $\overline{V}_h^k$, $\overline{N}_h^k$. We denote by $\pi_k$ and $\overline{\pi}_k$ the greedy policy w.r.t. $Q_h^k$ and $\overline{Q}_h^k$. We further use superscript $(n)$ to denote related quantities for task $n$, e.g. $Q_h^{k(n)}$ and $Q_h^{\pi_k(n)}$.

**Theorem 1.** *There exists an absolute constant $c > 0$ such that, for all $p \in (0,1)$, if we choose $b_t = c\sqrt{H^3(\log(N) + \iota)/t}$, we have that with probability at least $1 - p$, it takes Alg. 1 at most*

$$O((\log N + \iota)H^5 SA/\varepsilon^2) \tag{3}$$

*exploration episodes to simutaneously return an $\varepsilon$-optimal policy $\pi^{(n)}$ for each of the $N$ tasks.*

*Proof Sketch:* The proof of theorem 1 relies on a key lemma, which relates the estimation error of the $Q$ function on each task, $(Q_h^{k(n)} - Q_h^{\pi_k(n)})$, to the estimation error on the pseudo-$Q$ function, $(\overline{Q}_h^k - \overline{Q}_h^{\overline{\pi}_k})$. Note that $\overline{Q}$ corresponds to a zero-reward MDP, and thus $\overline{Q}_h^{\overline{\pi}_k} = 0$.

**Lemma 2.** *There exists an absolute constant $c > 0$ such that, for any $p \in (0,1)$, if we choose $b_t = c\sqrt{H^3(\log(N) + \iota)/t}$, we have that with probability at least $1 - p$, the following holds simultaneously for all $(s, a, h, k, n) \in S \times A \times [H] \times [K] \times [N]$:*

$$(Q_h^{k(n)} - Q_h^{\pi_k(n)})(s, a) \leq (\overline{Q}_h^k - \overline{Q}_h^{\overline{\pi}_k})(s, a). \tag{4}$$

Lemma 2 shows that the task Q function $Q_h^{k(n)}$ converges at least as fast as $\overline{Q}_h^k$. We then have

$$
(V_1^{*(n)} - V_1^{\pi_k(n)})(s_1) \overset{①}{\leq} \left( V_h^{k(n)} - V_h^{\pi_k(n)} \right)(s_1) \leq \max_a \left( Q_h^{k(n)} - Q_h^{\pi_k(n)} \right)(s_1, a)
$$

$$
\overset{②}{\leq} \max_a \left( \overline{Q}_h^k - \overline{Q}_h^{\overline{\pi}_k} \right)(s_1, a) \overset{③}{=} \left( \overline{V}_h^k - \overline{V}_h^{\overline{\pi}_k} \right)(s_1)
$$

where ① is due to [Jin et al., 2018, Lemma 4.3], ② is due to Lemma 2 and ③ is due to $\overline{Q}_h^{\overline{\pi}_k} = 0$. Since UCBZERO is mathematically equivalent to a zero-reward version of the UCB-H algorithm, we get $\sum_{k=1}^K (V_1^{*(n)} - V_1^{\pi_k(n)})(s_1) \leq \sum_{k=1}^K (\overline{V}_1^k - \overline{V}_1^{\overline{\pi}_k})(s_1) \leq \sqrt{(\log N + \iota)H^5 SAK}$, where the last step is due to [Jin et al., 2018, Theorem 1]. For each task $n$, define $\pi$ to be the non-stationary stochastic policy which uniformly sample a policy from $\pi_1, ..., \pi_K$. Then, $(V_1^{*(n)} - V_1^{\pi(n)})(s_1) = \left[ \sum_{k=1}^K (V_1^{*(n)} - V_1^{\pi_k(n)})(s_1) \right]/K \leq O\left( \sqrt{(\log N + \iota)H^5 SA/K} \right)$. This implies that in order for $(V_1^{*(n)} - V_1^{\pi(n)})(s_1) \leq \varepsilon$, we need at most $K = O\left( (\log N + \iota)H^5 SA/\varepsilon^2 \right)$. ∎

Theorem 1 indicates that, when $N$ is small, e.g. $N \leq O(\text{poly}(H, S, A))$, the sample complexity of UCBZERO to simultaneously learn $N$ tasks is **the same** in the leading terms as the sample complexity of UCB-H to learn a single task, despite that UCBZERO does not have access to rewards during exploration. In contrast, a naive application of UCB-H to learn $N$ tasks will require $N$ times the original sample complexity. This exponential improvement is achieved because UCBZERO is able to collect transitions in a way that helps with learning all tasks during a single exploration phase. On the other hand, the data collected by UCB-H guided by a specific task may under-explore certain regions of the environment that are not important to the current task, but may be important to other tasks. Even for the task-specific RL setting, our result implies that, perhaps surprisingly, a full-exploration algorithm can learn as fast as an algorithm that balances exploration vs. exploitation. It suggests that exploitation doesn't necessarily help in accelerating learning, and is probably only required for the sake of regret minimization.

## 4.2 Further Insights on the Exploration Phase of UCBZERO

In Theorem 1 above, we directly analyzed the sample complexity of UCBZERO, but the proof technique provides limited insight in the quality of the dataset $\mathcal{D}$ collected during the exploration phase. Intuitively, $\mathcal{D}$ must cover all reachable states sufficiently often to ensure that one can later learn near-optimal policy for any unknown reward functions. Our next result shows that this is indeed the case for UCBZERO.

**Definition 1.** *We define the relative reachability between $(s', h')$ and $(s, h)$, $h' < h$, by the maximum probability of reaching $(s, h)$ from $(s', h')$ following some policy. Specifically,*

$$\delta_{h',h}(s', s) = \max_{\pi} P^{\pi}(s_h = s | s_{h'} = s') \tag{5}$$

*We also define the **reachability** of $(s, h)$ by the distance from $(s_1, 1)$ to $(s, h)$, i.e. $\delta_h(s) = \delta_{1,h}(s_1, s)$.*

Intuitively, $\delta_h(s)$ represents the maximum probability of reaching state $s$ in step $h$. If $\delta_h(s)$ is zero or close to zero for some $(s, h)$, it means that it's almost impossible to reach state $s$ in step $h$, and thus $(s, h)$ will not have too much impact in the optimal performance of any task. Our next theorem shows that UCBZERO is able to consistently visit *all* state-action pairs during the whole exploration phase.

**Theorem 3.** *With probability $1 - p$, after $K$ episode of UCBZERO, we have*

$$N_h^K(s, a) \geq \Omega \left( \frac{K \delta_h(s)^2}{H^2 SA} \right). \tag{6}$$

*for all $(s, a, h) \in S \times A \times [H]$.*

Theorem 3 shows that UCBZERO will visit all state-action pairs in proportion to the total number of episodes, and the visitation frequency scales at most quadratically with the reachability of the state. One direct implication of sufficient visitation is that a model-based batch-RL algorithm, such as Tabular Certainty Equivalence (TCE) [Jiang, 2018], can make accurate estimations on the transition dynamics and thus find near-optimal policies for any reward functions:

**Theorem 4.** *With probability $1 - p$, for all $(h, s, a, s') \in [H] \times S \times A \times S$, Alg. 1 induces an estimate $\hat{P}_h(s'|s, a)$ of $P_h(s'|s, a)$, such that $|\hat{P}_h(s'|s, a) - P_h(s'|s, a)| \leq \frac{\varepsilon}{\delta_h(s)}$, after at most $O(H^5 SA\iota/\varepsilon^2)$ exploration episodes.*

These insights suggest that the efficiency of UCBZERO is likely not a consequence of careful cooperation between the exploration component and the policy optimization component, both using Q-learning with UCB bonuses. Instead, any batch RL algorithm can be applied in the policy optimization phase to find near-optimal policies.

## 5 Lower bound

In the standard task-specific RL setting, it is a common belief that having to estimate reward functions is not a statistical bottleneck to efficient learning, due to the relatively small number of parameters of a reward function compared to the number of parameters of the transition dynamics. However, we show that in the task-agnostic RL setting, the additional $\log N$ factor in the sample complexity of UCBZERO is unavoidable, and this additional complexity is essentially due to the need to accurately estimate $N$ reward functions simultaneously.

**Theorem 5.** *Any PAC-MDP algorithm in the task-agnostic RL setting must spend at least $\Omega \left( \log(N) H^2 SA/\varepsilon^2 \right)$ episodes in the exploration phase.*

*Proof Sketch:* Our main intuition is that when $N$ becomes very large, the sample complexity of simultaneously estimating $N$ reward functions eventually out-scales the sample complexity of estimating the shared transition dynamics. As a result, one can equivalently assume that the transition dynamics are already known to the learner, and therefore solving the MDP with a particular reward function becomes equivalent to solving $SH$ parallel multi-armed bandits each with $A$ arms.

Based on the this intuition, we construct an MDP $M$ where the transition is defined as $P_h(s'|s, a) = 1/S$ for all $(h, s, a, s')$ and **is known** to the learner. Since the action has no control over the next-state, this is equivalent to a collection of $SH$ multi-armed bandits. Due to the uniform transition,

$P_h^\pi(s) = 1/S$ for any $(\pi, s, h)$, and so finding the $\varepsilon$-optimal policy amounts to finding an $\varepsilon/H$-optimal policy for each bandit $(s, h)$. We then construct each bandit following the classic lower bound construction of multi-armed bandits [Mannor and Tsitsiklis, 2004], where the reward for all arms are Bernoulli with $p = 1/2$, except for the first arm that has Bernoulli with $p = 1/2 + \varepsilon/2$ and one other arm that has Bernoulli $p = 1/2 + \varepsilon$. The classic result shows that such a bandit requires at least $\Omega(AH^2/\varepsilon^2)$ steps to find an $\varepsilon/H$-optimal arm. We extend it and show that it takes at least $\Omega(\log(N)H^2A/\varepsilon^2)$ steps to find an $\varepsilon/H$-optimal arm simultaneously for $N$ arbitrary reward functions in the *task-agnostic bandit* setting. As a result, the $HS$ bandits requires at least $\Omega(\log(N)H^3SA/\varepsilon^2)$ steps, which can be achieved in at least $\Omega(\log(N)H^2SA/\varepsilon^2)$ episodes. ∎

The lower bound suggest that our bound achieved by UCBZERO is tight in $S, A, \varepsilon$, up to logarithmic factors and lower-order terms. In the next section, we discuss the connections of our upper and lower bound results to the ones in the standard task-specific RL setting and the reward-free setting.

## 6 Task-Agnostic vs. Reward-free RL

In this section, we (1) make a thorough comparison with the comtemporary work [Jin et al., 2020], (2) provide additional results on the sample complexity of UCBZERO in the reward-free setting, and (3) present a unified view of the three RL frameworks. A summary is presented in Table 1.

### 6.1 Summary of the Reward-free RL setting

A comtemporary work [Jin et al., 2020] proposed the *reward-free RL* framework that is similar to our task-agnostic RL framework, where learning is decomposed into the exploration phase and the planning phase. In the exploration phase, the learner explores without reward information. In the planning phase, the agent is tasked with computing near-optimal policies for a large collection of **given reward functions**. The only essential difference between the two frameworks is whether true reward functions are known (reward-free), or only instantiated rewards are available (task-agnostic). Therefore, our task-agnostic setting is statistically harder than the reward-free setting.

Under the reward-free setting, Jin et al. focus on providing $N$-independent bounds on the sample complexity. Intuitively, this is possible since the only unknown quantity in the problem is the transition dynamics that are shared across all tasks. Therefore, as long as the algorithm achieves an accurate estimate of the transition dynamics, it can compute near-optimal policies under arbitrary known reward functions. In the $N$-independent regime, Jin et al. provided an $\Omega\left(H^2S^2A/\varepsilon^2\right)$ lower bound, which has an **additional factor of S** compared to the lower bound in the standard task-specific RL setting. The lower bound construction requires an exponential number of rewards, i.e. $N \geq \exp(S)$, and they emphasized that it remains an **open problem** whether smaller $N$-dependent bound is possible when $N$ is finite.

In addition, they presented an efficient algorithm RFE that has sample complexity $\tilde{O}\left(H^5S^2A/\varepsilon^2\right)$, matching the lower bound in the dependency on $S, A$, and $\varepsilon$. During exploration, the RFE algorithm executes two meta-steps. In the first step, the algorithm calls a SOTA PAC-MDP algorithm, EULER [Zanette and Brunskill, 2019], as a subroutine for $HS$ times to learn a navigation policy $\pi_{h,s}$ that visit state $s$ in step $h$ as often as possible. In the second step, RFE collects data with randomly sampled policies from $\{\pi_{h,s}\}$ to ensure that all $(h, s)$ pairs can be visited sufficiently often. While the idea is conceptually simple, the algorithm can be expensive to execute in practice, as the overhead of calling EULER $HS$ times can already be large. In fact, our result indicates that calling UCBZERO once suffices to learn all navigation policies $\{\pi_{h,s}\}$ with only an additional $\log(HS)$ factor more samples, as opposed to $HS$ times more by calling EULER $HS$ times.

### 6.2 UCBZERO in Reward-free RL

Since our task-agnostic setting is strictly more difficult than the reward-free setting, the $\tilde{O}(\log(N)H^5SA/\varepsilon^2)$ upper bound readily applies to the reward-free setting. Our result immediately implies that a **tighter N-dependent bound** is available in the reward-free setting, answering the open question raised by Jin et al. Compared with the bound achieved by RFE, our bound replaces the extra dependency on $S$ with a logarithmic dependency on $N$. When the number of tasks is not extremely large, e.g. $N \leq O(\text{poly}(H, S, A))$, UCBZERO achieves a tighter bound than RFE.

In the rare case that $N$ is exponentially large or even infinite, the bound in theorem 1 will blow up. Our next result complements theorem 1 by providing an $N$-independent bound on the sample-complexity of UCBZERO in the reward-free setting, with the price of an additional factor of $HSA$.

**Theorem 6.** *There exists an absolute constant $c > 0$ such that, for all $p \in (0, 1)$, we have that with probability at least $1 - p$, it takes Alg. 1 at most*

$$O\left(H^6 S^2 A^2 (\log \frac{3H}{\varepsilon} + \iota)/\varepsilon^2\right) \tag{7}$$

*exploration episodes to simutaneously return an $\varepsilon$-optimal policy $\pi$ for any number of tasks.*

*Proof Sketch:* The proof is based on the observation that if two reward functions are close enough, e.g. $|\mathbf{E}\left[r_h(s, a)\right] - \mathbf{E}\left[r'_h(s, a)\right]| \leq \varepsilon$ for all $(s, a, h)$, then a near-optimal policy for $r$ will also be near-optimal for $r'$. This is sometimes referred as the Simulation Lemma [Kearns and Singh, 2002]. As a result, even though there are potentially infinite number of reward functions to optimize, we can divide the whole space of rewards $[0, 1]^{H,S,A}$ into $M^{HSA}$ disjoint sets of the form $\Pi_{h,s,a}[\frac{i_{h,s,a}-1}{M}, \frac{i_{h,s,a}}{M}]$. For sufficiently large $M$, one only needs to find a near-optimal policy for one reward function per set, to guarantee that this policy will be near-optimal for all reward functions in the same set. Therefore, we effectively only need to find a near-optimal policy for at most $N = M^{HSA}$ tasks. Plugging this $N$ into theorem 1 gives the desired result. ∎

This $N$-independent bound is not as sharp as the bound for RFE, likely due to the simplicity of the argument we use. Nevertheless, the key takeaway is that the sample complexity of UCBZERO will not scale with $N$ indefinitely. In most practical scenarios, the number of tasks to be learned is small. For example, in the navigation problem, the agent wants to learn to navigate to all $(s, h) \in \mathcal{S} \times [H]$. This implies a task number of $N = SH$. Therefore, the bound achieved by UCBZERO is usually tighter in practice. What remains an **open question** is whether a smaller $N$-dependent lower bound exist in the reward-free setting. We note that our lower bound for the task-agnostic setting does not readily transfer to the easier reward-free setting.

## 6.3 A Unified View

Lastly, we try to provide a unified view of the three frameworks: the standard task-specific RL, the reward-free RL, and the task-agnostic RL. An overview of technical results in each setting is presented in Table 1. The task-specific RL aims at learning near-optimal policy for **a single task**, with or without knowledge of the true reward function. The reward-free RL aims at learning near-optimal policy for a set of tasks, **given** the knowledge of the true reward functions. The task-agnostic RL aims at learning near-optimal policy for a set of tasks, but **without** knowledge of the true reward functions. In terms of statistical difficulty, there is a total order between the three, namely "task-specific RL < reward-free RL < task-agnostic RL". Task-specific RL is well-understood, with matching $\Theta(H^2 SA/\varepsilon^2)$ upper and lower bound. Reward-free RL is not fully-solved, as there are both $N$ dependent bounds and $N$-independent bounds. The additional dependency on $S$ is proved unavoidable in an $N$-independent bound but seems to be avoidable in an $N$-dependent one. It remains an open question whether there exists a tighter $N$-dependent lower bound. Task-agnostic RL is the hardest. UCBZERO enjoys the same $N$-dependent upper bound in both the reward-free and task-agnostic setting. Our $N$-dependent lower bound also suggests that $N$-independent bounds **do not exist** in the task-agnostic setting. Nevertheless, our upper and lower bounds do not yet match in the H factors. And it remains an open question whether zero-reward versions of other PAC-MDP algorithms in the task-specific RL setting can be applied to the task-agnostic RL setting to achieve a tighter bound.

## 7 Conclusions

In this paper, we propose a new *task-agnostic RL* framework, that consists of an exploration phase, where the learner collects data without reward information, and a policy-optimization phase, where the learner computes near-optimal policies for $N$ tasks given instantiated rewards. We present an efficient algorithm, UCBZERO that finds $\varepsilon$-optimal policies on $N$ tasks within $\tilde{O}(\log(N)H^5 SA/\varepsilon^2)$ exploration episodes. We also provide a near-matching $\Omega(\log(N)H^2 SA/\varepsilon^2)$ lower bound, demonstrating the near-optimality of our algorithm in this setting.

## Broader Impact

Our work provides theoretical foundation for empirical studies of multi-task reinforcement learning and unsupervised reinforcement learning.

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
