[Supplementary Material]

# Appendices

## A   Proof of Upper Bound

In the proofs below we drop the superscript $(n)$ for simplicity.

***Proof of Lemma 2***.  Similar to Lemma 4.2 of [Jin et al., 2018], we can write down a recursive formula for both $(Q_h^k - Q_h^{\pi_k})(s, a)$ and $(\overline{Q}_h^k - \overline{Q}_h^{\overline{\pi}_k})(s, a)$, and perform a subtraction, which gives

$$[(\overline{Q}_h^k - \overline{Q}_h^{\overline{\pi}_k}) - (Q_h^k - Q_h^{\pi_k})](s, a) \tag{8}$$

$$= \alpha_t^0 \left( Q_h^{\pi_k} - \overline{Q}_h^{\overline{\pi}_k} \right)(s, a) \tag{9}$$

$$+ \sum_{i=1}^t \alpha_t^i \left[ (\overline{V}_{h+1}^{k_i} - \overline{V}_{h+1}^{\overline{\pi}_k}) - (V_{h+1}^{k_i} - V_{h+1}^{\pi_k}) \right] (s_{h+1}^{k_i}) \tag{10}$$

$$+ \sum_{i=1}^t \alpha_t^i \left[ [\hat{\mathbb{P}}^{k_i} - \mathbb{P}_h][\overline{V}_{h+1}^{\overline{\pi}_k} - V_{h+1}^{\pi_k}](s, a) + (r_h^{k_i} - \mathbf{E}\left[r_h\right](s_h^{k_i}, s_h^{k_i})) \right] \tag{11}$$

$$+ \sum_{i=1}^t \alpha_t^i b_i. \tag{12}$$

We now show that $[(\overline{Q}_h^k - \overline{Q}_h^{\overline{\pi}_k}) - (Q_h^k - Q_h^{\pi_k})](s, a) \geq 0$ by induction on $h = H, H - 1, ..., 1$ and $k = 1, ..., K$. It is easy to see that the first term of right-hand side $\alpha_t^0 \left( Q_h^{\pi_k} - \overline{Q}_h^{\overline{\pi}_k} \right)(s, a) \geq 0$ since $\overline{Q}_h^{\overline{\pi}_k} = 0$. For the second term, consider two cases:

(1) If $\max_a \overline{Q}_{h+1}^{k_i}(s_{h+1}^{k_i}, a) \geq H$, then $\left[ (\overline{V}_{h+1}^{k_i} - \overline{V}_{h+1}^{\overline{\pi}_k}) - (V_{h+1}^{k_i} - V_{h+1}^{\pi_k}) \right] (s_{h+1}^{k_i}) = H - (V_{h+1}^{k_i} - V_{h+1}^{\pi_k})(s_{h+1}^{k_i}) \geq 0$.

(2) If $\max_a \overline{Q}_{h+1}^{k_i}(s_{h+1}^{k_i}, a) < H$, then

$$(\overline{V}_{h+1}^{k_i} - \overline{V}_{h+1}^{\overline{\pi}_{k_i}})(s_{h+1}^{k_i}) = \max_a \left[ \overline{Q}_{h+1}^{k_i}(s_{h+1}^{k_i}, a) - \overline{Q}_{h+1}^{\overline{\pi}_{k_i}}(s_{h+1}^{k_i}, a) \right] \tag{13}$$

$$\geq \max_a \left[ Q_{h+1}^{k_i}(s_{h+1}^{k_i}, a) - Q_{h+1}^{\pi_{k_i}}(s_{h+1}^{k_i}, a) \right] \tag{14}$$

$$\geq Q_{h+1}^{k_i}(s_{h+1}^{k_i}, \pi_{k_i}(s_{h+1}^{k_i})) - Q_{h+1}^{\pi_{k_i}}(s_{h+1}^{k_i}, \pi_{k_i}(s_{h+1}^{k_i})) \tag{15}$$

$$= (V_{h+1}^{k_i} - V_{h+1}^{\pi_{k_i}})(s_{h+1}^{k_i}) \tag{16}$$

where the first inequality is by the induction hypothesis.

Similar to the proof of Lemma 4.3 in [Jin et al., 2018], observe that $(\mathbb{1}\left[k_i \leq K\right] \cdot \left[ [\hat{\mathbb{P}}^{k_i} - \mathbb{P}_h][\overline{V}_{h+1}^{\overline{\pi}_k} - V_{h+1}^{\pi_k}](s, a) + (r_h^{k_i} - \mathbf{E}\left[r_h\right](s_h^{k_i}, s_h^{k_i})) \right])_{i=1}^\tau$ is a martingale difference sequence. By Azuma-Hoeffding, we have that with probability $1 - p/(SAHN)$:

$$\forall \tau \in [K], \left| \sum_{i=1}^\tau \alpha_\tau^i (\mathbb{1}\left[k_i \leq K\right] \cdot \left[ [\hat{\mathbb{P}}^{k_i} - \mathbb{P}_h][\overline{V}_{h+1}^{\overline{\pi}_k} - V_{h+1}^{\pi_k}](s, a) + (r_h^{k_i} - \mathbf{E}\left[r_h\right](s_h^{k_i}, s_h^{k_i})) \right]) \right|$$

$$\leq \frac{c'(H + 1)}{2} \sqrt{\sum_{i=1}^\tau (\alpha_\tau^i)^2 \cdot (\log N + \iota)} \leq c \sqrt{\frac{H^3(\log N + \iota)}{\tau}},$$

for some absolute constant $c$. Using a union bound, we see that with at least probability $1 - p$, the following holds simultaneously for all $(s, a, h, k, n) \in S \times A \times [H] \times [K] \times [N]$:

$$\left| \sum_{i=1}^t \alpha_t^i (\mathbb{1}\left[k_i \leq K\right] \cdot [\hat{\mathbb{P}}^{k_i} - \mathbb{P}_h][\overline{V}_{h+1}^{\overline{\pi}_k} - V_{h+1}^{\pi_k}](s, a)) \right| \leq c \sqrt{\frac{H^3(\log N + \iota)}{t}} \tag{17}$$

Finally, since we choose $b_t = c\sqrt{\frac{H^3(\log N + \iota)}{t}}$, we have that the last two terms of $[(\overline{Q}_h^k - \overline{Q}_h^{\pi_k}) - (Q_h^k - Q_h^{\pi_k})](s,a)$ also adds up at least zero. Putting everything together, we have shown that with probability at least $1 - p$, $[(\overline{Q}_h^k - \overline{Q}_h^{\pi_k}) - (Q_h^k - Q_h^{\pi_k})](s,a) > 0$ for all $(s,a,h,k,n) \in S \times A \times [H] \times [K] \times [N]$. This concludes the proof. ∎

Given Lemma 2, the proof of Theorem 1 follows from the proof sketch in the main text.

# B  Proof of Additional Properties of UCBZERO

We first give two lemmas:

**Lemma 7.** *For any $(s,a,h,k) \in S \times A \times [H] \times [K]$, let $t = N_h^k(s,a)$, then we have*

$$\overline{V}_h^k(s) \geq \min(H, b_t) \tag{18}$$

*Proof.* We have, for any $(s,a,h,k) \in S \times A \times [H] \times [K]$,

$$\overline{Q}_h^k(s,a) = \alpha_t^0 H + \sum_{i=1}^{t} \alpha_t^i \left[ \overline{V}_{h+1}^{k_i}(x_{h+1}^{k_i}) + b_i \right] \tag{19}$$

$$\geq \sum_{i=1}^{t} \alpha_t^i b_i \geq \sum_{i=1}^{t} \alpha_t^i b_t = b_t. \tag{20}$$

Thus, $\overline{V}_h^k(s) \geq \min(H, \max_a \overline{Q}_h^k(s,a)) \geq \min(H, b_t)$. ∎

**Lemma 8.** *With probability at least $1 - p$, for any $(s,a,h,k) \in S \times A \times [H] \times [K]$, let $t = N_h^k(s,a)$, then we have*

$$\overline{Q}_h^k(s,a) \geq \sum_{i=1}^{t} \alpha_t^i [\mathbb{P}_h \overline{V}_{h+1}^{k_i}](s,a) \tag{21}$$

*Proof.* We have, for any $(s,a,h,k) \in S \times A \times [H] \times [K]$,

$$Q_h^k(s,a) = \alpha_t^0 H + \sum_{i=1}^{t} \alpha_t^i \left[ V_{h+1}^{k_i}(x_{h+1}^{k_i}) + b_i \right] \tag{22}$$

$$\geq \sum_{i=1}^{t} \alpha_t^i \left[ \mathbb{P}_h \overline{V}_{h+1}^{k_i}(s,a) + (\hat{\mathbb{P}}_h^{k_i} - \mathbb{P}_h)\overline{V}_{h+1}^{k_i}(s,a) + b_i \right] \tag{23}$$

$$\overset{\text{w.p. 1-p}}{\geq} \sum_{i=1}^{t} \alpha_t^i \left[ \mathbb{P}_h \overline{V}_{h+1}^{k_i}(s,a) \right] \tag{24}$$

The last inequality is by the same martingale bound as in the proof of Lemma 2. ∎

Now, we are ready to prove Theorem 3.

***Proof of Theorem 3:*** Let $h^*, s^*, a^*$ be given, and denote $t^* = N_{h^*}^K(s^*, a^*)$, $b_t^* = c\sqrt{H^3\iota/t^*}$. Then, by Lemma 7, we have

$$\overline{V}_{h^*}^K(s^*, a^*) \geq \min(H, b_t^*) \tag{25}$$

Now, by Lemma 8, we have that for any $(s,a)$,

$$\overline{Q}_{h^*-1}^K(s,a) \geq \sum_{i=1}^{t} \alpha_t^i [\mathbb{P}_{h^*-1} \overline{V}_{h^*}^{k_i}](s,a) \tag{26}$$

$$\geq \sum_{i=1}^{t} \alpha_t^i P(s^*|s,a) \min(H, b_i) \tag{27}$$

$$= P(s^*|s,a) \min(H, b_t^*) \tag{28}$$

Thus, we have

$$\overline{V}_{h^*-1}(s) \quad = \quad \max_a \overline{Q}_{h^*-1}(s,a) \tag{29}$$

$$\geq \quad \max_a P(s^*|s,a)\min(H,b_t^*) \tag{30}$$

$$= \quad \delta_{h^*-1,h^*}(s,s^*)\min(H,b_t^*) \tag{31}$$

We now show by induction that for all $h < h^*$,

$$\overline{V}_h(s) \geq \delta_{h,h^*}(s,s^*)\min(H,b_t^*) \tag{32}$$

We again use Lemma 8 to get

$$\overline{Q}_h(s,a) \quad \geq \quad \sum_{i=1}^{t} \alpha_t^i [\mathbb{P}_h \overline{V}_{h+1}^{k_i}](s,a) \tag{33}$$

$$\geq \quad \sum_{i=1}^{t} \alpha_t^i [\sum_{s'\in S} P(s'|s,a)\overline{V}_{h+1}^{k_i}(s')] \tag{34}$$

$$\geq \quad \sum_{i=1}^{t} \alpha_t^i [\sum_{s'\in S} P(s'|s,a)\delta_{h+1,h^*}(s',s^*)\min(H,b_t^*)] \tag{35}$$

$$= \quad \sum_{s'\in S} P(s'|s,a)\delta_{h+1,h^*}(s',s^*)\min(H,b_t^*) \tag{36}$$

Then,

$$\overline{V}_h(s) \quad = \quad \max_a \sum_{s'\in S} P(s'|s,a)\delta_{h+1,h^*}(s',s^*)\min(H,b_t^*) \tag{37}$$

$$= \quad \delta_{h,h^*}(s',s^*)\min(H,b_t^*), \tag{38}$$

where in the last equality, we use the Bellman optimality equation w.r.t. $\delta$, i.e. $\delta_{h,h^*}(s',s^*) = \max_a \sum_{s'\in S} P(s'|s,a)\delta_{h+1,h^*}(s',s^*)$. Therefore, we have established that $\overline{V}_1(s) \geq \delta_{1,h^*}(s,s^*)\min(H,b_t^*)$. This implies that

$$\sum_{k=1}^{K} \overline{V}_1^k(s_1) \geq K\delta(s^*)\min(H,b_t^*) \tag{39}$$

Furthermore, we know from the proof of Theorem 1 that

$$\sum_{k=1}^{K} \overline{V}_1^k(s_1) \leq O(\sqrt{H^5 SAK\iota}) \tag{40}$$

When $K \geq \Omega\left(\frac{H^3 SA\iota}{\delta(s^*)}\right)$, $\sum_{k=1}^{K} \overline{V}_1^k(s_1) \leq K\delta(s^*)H$. Therefore, we have

$$K\delta(s^*)b_t^* = K\delta(s^*)c\sqrt{H^3\iota/t^*} \leq O(\sqrt{H^5 SAK\iota}) \tag{41}$$

This gives us

$$t^* \geq O(\frac{K\delta(s^*)^2}{H^2 SA}). \tag{42}$$

This holds for any $s^*, a^*, h^*$, establishing the results. ∎

***Proof of Theorem 4.*** First, notice that for any given $r_h(s,a,s')$ out of a set of size $HS^2A$, by the proof of Theorem 1, we have

$$\sum_{k=1}^{K}(V_1^k - V_1^*)(s_1) \leq \sum_{k=1}^{K} \overline{V}_1^k \leq O\left(\sqrt{H^5 SA\iota K}\right) \tag{43}$$

Define $\tilde{V}_1^K = \frac{1}{K}\sum_{k=1}^K V_1^k(s_1)$, then

$$0 \le \tilde{V}_1^K - V_1^*(s_1) \le O\left(\sqrt{\frac{H^5 SA\iota}{K}}\right) \le \varepsilon. \tag{44}$$

Now, let $(h^*, s^*, a^*, s'^*)$ be given. Define reward functions $R^{(1)}, R^{(2)}$ as

$$R_h^{(1)}(s, a, s') = \begin{cases} 1, & \text{if } h = h^*, s = s^*, a = a^*, s' = s'^* \\ 0, & \text{otherwise} \end{cases} \tag{45}$$

$$R_h^{(2)}(s, a, s') = \begin{cases} 1, & \text{if } h = h^*, s = s^* \\ 0, & \text{otherwise} \end{cases} \tag{46}$$

Then, we observe that the corresponding $V_1^{*(1)} = \delta_{h^*}(s^*)P_{h^*}(s'^*|s^*, a^*)$ and $V_1^{*(2)} = \delta_{h^*}(s^*)$. Now, define

$$\hat{P}(s'^*|s^*, a^*) = \frac{\tilde{V}_1^{K(1)}}{\tilde{V}_1^{K(2)}} \tag{47}$$

Next, we show that $\|\hat{P}(s'^*|s^*, a^*) - P(s'^*|s^*, a^*)\|$ is small. In particular,

$$\hat{P}(s'^*|s^*, a^*) = \frac{\tilde{V}_1^{K(1)}}{\tilde{V}_1^{K(2)}} \tag{48}$$

$$\le \frac{\delta_{h^*}(s^*)P_{h^*}(s'^*|s^*, a^*) + \varepsilon}{\delta_{h^*}(s^*)} \tag{49}$$

$$= P(s'^*|s^*, a^*) + \frac{\varepsilon}{\delta_{h^*}(s^*)}, \tag{50}$$

$$\hat{P}(s'^*|s^*, a^*) = \frac{\tilde{V}_1^{K(1)}}{\tilde{V}_1^{K(2)}} \tag{51}$$

$$\ge \frac{\delta_{h^*}(s^*)P_{h^*}(s'^*|s^*, a^*)}{\delta_{h^*}(s^*) + \varepsilon} \tag{52}$$

$$\ge \frac{\delta_{h^*}(s^*)P_{h^*}(s'^*|s^*, a^*) - \varepsilon}{\delta_{h^*}(s^*)} \tag{53}$$

$$= P(s'^*|s^*, a^*) - \frac{\varepsilon}{\delta_{h^*}(s^*)}. \tag{54}$$

A union bound on all $(h^*, s^*, a^*, s'^*) \in [H] \times S \times A \times S$ completes the proof. Notice that the sample complexity only changes by constant factor as $\log(N) = \log(HS^2 A) \le 2\log(HSA)$. ∎

## C   Proof of Lower Bound

We based our construction on the classic lower-bound construction for multi-armed bandits. For a detailed introduction of the problem setting, please refer to [Mannor and Tsitsiklis, 2004]. We first introduce some bandit notation: let $n$ be the number of arms, $p \in [0, 1]^n$ represent the parameters of the Bernoulli distribution of rewards associated with each arm. We let $T_\ell$ be the total number of times that arm $\ell$ is pulled, and $T = \sum_{\ell=1}^n T_\ell$ be the total number of arm pulls. We also let $I$ be the arm that is selected at the end of the exploration phase.

**Lemma 9.** *There exists a $p \in [0, 1]^n$, $n \ge 2$ such that for any fixed number of episodes $K$, there exists $N = O(2^K)$ reward functions, so that with probability at least 0.5, no RL algorithm can learn an $\varepsilon$-optimal policy with $\varepsilon \le 0.08$ for at least one reward function.*

*Proof.* We construct a bandit with two arms $\ell = 1, 2$. We consider two reward functions. The first reward function is $p$ with $p_1 = 0.1, p_2 = 0$ and the second reward function is $q$ with $q_1 = 0.1$, $q_2 = Bernoulli(0.5)$. Thus, it is easy to see that the optimal arm corresponding to $p$ and $q$ are $\ell = 1$ and $\ell = 2$ respectively. We assume among the $N$ reward functions we need to learn, $N - 1$ of them

are $q$ and only one is $p$. Next, we show that no learner is able to distinguish whether the instantiated rewards are from $p$ or $q$.

Let $T_2$ be the number of episodes where arm 2 is taken in the $K$ instantiated rewards. Then for each of the $N-1$ reward function $q$, it has probability $0.5^{T_2}$ to generate the same instantiated rewards with $r_1$. Note that $0.5^{T_2} \geq 0.5^K$, so the probability that at least one of the $q$ generate the same instantiated rewards as $p$ is at least

$$1 - (1 - 0.5^K)^{N-1} \geq 1 - e^{-0.5^K(N-1)} \tag{55}$$

Let $N = \lceil 1 + 2^K \ln 2 \rceil$, then the probability that the rewards can be generated by one of the $q$ is at least $0.5$. Given such a reward configuration, let $\hat{\pi} = (x, 1-x)$ be the learned (stochastic) policy where $x$ is the probability of choosing arm $1$. Then for reward function $q$, the optimality gap is

$$V_2^* - V_2(\hat{\pi}) = 0.5 - 0.1x - (1-x)*0.5 = 0.4x, \tag{56}$$

while for reward function $r_1$, the optimality gap is

$$V_1^* - V_1(\hat{\pi}) = 0.1 - 0.1x. \tag{57}$$

One can see that regardless of $p_1$, one of the above two gaps will be large, and the minimum of $\max(V_2^* - V_2(\hat{\pi}), V_1^* - V_1(\hat{\pi}))$ is achieved when $p_1 = 0.2$, and the minimum value is $0.08$.

Therefore with probability at least $0.5$, no RL algorithm can learn $\varepsilon$-optimal policy with $\varepsilon = 0.08$. ∎

**Theorem 10.** *There exist some positive constant $c_1$, $c_2$, $\varepsilon_0$, $\delta_0$, such that for every $n \geq 2$, $\varepsilon \in (0, \varepsilon_0)$, and $\delta \in (0, \delta_0)$, and for every $(\varepsilon, \delta)$-correct policy on $N$ tasks, there exists some $p \in [0,1]^n$ such that*

$$\mathbf{E}_p[T] \geq c_1 \frac{n}{\varepsilon^2} \log \frac{c_2 N}{\delta} \tag{58}$$

*Proof.* The proof largely mimic the original proof of Theorem 1 in [Mannor and Tsitsiklis, 2004], with the distinction in handling $N$ tasks instead of 1. Consider a bandit problem with $n+1$ arms. We also consider a finite set of $n+1$ possible reward functions $p$, which we refer to as "hypotheses". Under any one of the hypothesis, arm $0$ has a Bernoulli reward with $p_0 = (1+\varepsilon)/2$. Under one hypothesis, denoted $H_0$, all other arm has $p_i = 1/2$, which makes arm $0$ the best arm. Furthermore, for $\ell = 1, ..., n$, there is a hypothesis

$$H_\ell : p_0 = \frac{1+\varepsilon}{2}, \quad p_\ell = \frac{1}{2} + \varepsilon, \quad p_i = \frac{1}{2}, \text{for } i \neq 0, \ell. \tag{59}$$

which makes arm $\ell$ the best arm. We define $\varepsilon_0 = 1/8$ and $\delta_0 = e^{-4}/8$. From now on, we fix $\varepsilon \in (0, \varepsilon_0)$, $\delta \in (0, \delta_0)$, $N \geq 1$ and a policy, which we assume to be $(\varepsilon/2, \delta)$-correct on $N$ rewards. If $H_0$ is true, the policy must have a probability at least $1 - \delta$ of eventually stopping and selecting arm $0$. If $H_\ell$ is true, for some $\ell \neq 0$, the policy must have a probability at least $1 - \delta$ of eventually stopping and selecting arm $\ell$. These further hold simultaneously for $N$ hypotheses. We denote $P_\ell^N(\cdot)$ as the probability of some event that happens simultaneously under $N$ $H_\ell$ hypotheses.

We define $t^*$ by

$$t^* = \frac{1}{c\varepsilon^2} \log \frac{N}{8\delta} = \frac{1}{c\varepsilon^2} \log \frac{N}{\theta} \tag{60}$$

where $\theta = 8\delta$ and $c$ is an absolute constant we will specify later. Note that $\theta < e^{-4}$ and $\varepsilon \leq 1/4$.

We assume by contradtion that $\mathbf{E}[T_1] \leq t^*$. We will eventually show that under this assumption, the probability of selecting $H_0$ under one of $N$ $H_1$ exceeds $\delta$, thus violates $(\varepsilon/2, \delta)$-correctness.

We now introduce some special events A, B and C. We define

$$A = \{T_1 \leq 4t^*\} \tag{61}$$

$$B = \{I = 0, \text{i.e. the policy eventually pick arm } 0\} \tag{62}$$

$$C = \left\{ \max_{1 \leq t \leq 4t^*} |K_t - \frac{1}{2}t| < \sqrt{t^* \log(N/\theta)} \right\} \tag{63}$$

where $K_t$ is the number of getting reward $1$ if the first arm is pulled $t$ times. Similar to the original proof [Mannor and Tsitsiklis, 2004], we have the following lemmas.

**Lemma 11.** $P_0^N(A) = P_0(A) > 3/4$, where $P_0^N(C)$ denotes the probability of event $B$ under all of $N$ hypothesis $H_0$.

This is directly due to the definition of $A$ that is independent of rewards and the use of Markov inequality.

**Lemma 12.** $P_0^N(B) > 3/4$.

This is due to $\delta < e^{-4}/8 < 1/4$.

**Lemma 13.** $P_0^N(C) > 3/4$.

This is due to the observation that $K_t - t/2$ is a martingale, and by applying Kolmogorov's inequality.

**Lemma 14.** If $0 \leq x \leq 1$ and $y \geq 0$, then

$$(1 - x)^y \geq e^{-dxy} \tag{64}$$

where $d = 1.78$

This is straightforward arithmetics. Please refer to the original proof in [Mannor and Tsitsiklis, 2004] for the detailed proofs of the lemmas. Let $S = A \cap B \cap C$, then we have $P_0^N(S) > 1/4$. Now we are ready to prove our main results. Let $W$ be the history of the process (the number of arm pulls for each arm in the exploration phase, and the sampled rewards in the policy-optimization phase). We define $L_\ell(W)$ to be the likelihood of a history $W$ under reward function $\ell$. We denote $K$ be a shorthand notation for $K_{T_1}$, the number of reward 1 instantiated on arm $\ell = 1$. Observe that, given the history up to time $t - 1$, the arm choice at time $t$ has the same probability distribution under either hypothesis $H_0$ and $H_1$; similarly, the arm reward at time $t$ has the same probability distribution, under either hypothesis, unless the chosen arm was arm 1. For this reason, the likelihood ratio $L_1(W)/L_0(W)$ is given by

$$\frac{L_1(W)}{L_0(W)} = \frac{(\frac{1}{2} + \varepsilon)^K (\frac{1}{2} - \varepsilon)^{T_1 - K}}{(\frac{1}{2})^{T_1}} \tag{65}$$

$$= (1 - 4\varepsilon^2)^K (1 - 2\varepsilon)^{T_1 - 2K} \tag{66}$$

Let $T_1^N(W)$ be the likelihood that $W$ appears under one of $N$ hypothese $H_1$. Since the instantiation of rewards under each hypothesis is completely independent from one another, we have

$$L_1^N(W) = 1 - (1 - L_1(W))^N \tag{67}$$

$$\geq 1 - \frac{1}{1 + L_1(W)N} \tag{68}$$

$$= \frac{L_1(W)N}{1 + L_1(W)N} \tag{69}$$

By lemma 9, we have that in order for the policy to be $\varepsilon, \delta$-correct, $T_1 \geq \log_2(N)$. Thus, we have

$$L_1(W) \leq (\frac{1}{2} + \varepsilon)^K (\frac{1}{2} - \varepsilon)^{T_1 - K} \tag{70}$$

$$\leq (\frac{1}{2})^{T_1} \tag{71}$$

$$\leq \frac{1}{N} \tag{72}$$

We then have

$$\frac{L_1^N(W)}{L_0(W)} = \frac{L_1(W)N}{1 + L_1(W)N} \frac{1}{L_0(W)} \tag{73}$$

$$\geq \frac{N}{2} \frac{L_1(W)}{L_0(W)} \tag{74}$$

$$= \frac{N}{2}(1 - 4\varepsilon^2)^K (1 - 2\varepsilon)^{T_1 - 2K} \tag{75}$$

If event $S$ occurred, then $A$ occurred, and we have $K \le T_1 \le 4t^*$, so that

$$(1 - 4\varepsilon^2)^K \ge (1 - 4\varepsilon^2)^{4t^*} = (1 - 4\varepsilon^2)^{\frac{1}{c\varepsilon^2} \log \frac{N}{\theta}} \qquad (76)$$

$$\ge e^{-(16d/c) \log(N/\theta)} \qquad (77)$$

$$= (\theta/N)^{16d/c} \qquad (78)$$

We have used here Lemma 14, which applies because $4\varepsilon^2 < 4/4^2 < 1/\sqrt{2}$. Similarly, if event S has occurred, then $A \cap C$ has occurred, which implies

$$T_1 - K \le 2\sqrt{t^* \log(N/\theta)} = (2/\varepsilon\sqrt{c}) \log(N/\theta). \qquad (79)$$

Therefore,

$$(1 - 2\varepsilon)^{T1 - 2K} \ge (1 - 2\varepsilon)^{(2/\varepsilon\sqrt{c}) \log(N/\theta)} \qquad (80)$$

$$\ge e^{-(4d/\sqrt{c} \log(N/\theta))} \qquad (81)$$

$$= (\theta/N)^{4d/\sqrt{c}} \qquad (82)$$

Substituting the above into the main equation, we obtain

$$\frac{L_1^N(W)}{L_0(W)} \ge \frac{N}{2}(\theta/N)^{(16d/c) + 4d/\sqrt{c}} \qquad (83)$$

By picking $c$ large enough ($c = 100$ suffices), we obtain that $\frac{L_1^N(W)}{L_0(W)} \ge \theta/2 \ge 4\delta$ whenever the event $S$ occurs. More precisely, we have

$$\frac{L_1^N(W)}{L_0(W)} \mathbb{1}[S] \ge 4\delta \mathbb{1}[S] \qquad (84)$$

where $\mathbb{1}[S]$ iss the indicator function of the event $S$. Then,

$$P_1^N(B) \ge P_1^N(S) = \mathbb{E}_1^N[\mathbb{1}[S]] = \mathbb{E}_0^N[\frac{L_1^N(W)}{L_0(W)} \mathbb{1}[S]] \ge \mathbb{E}_0^N[4\delta \mathbb{1}[S]] = 4\delta P_0^N(S) > \delta. \qquad (85)$$

where we used the fact that $P_0^N(S) > 1/4$. This contradict the assumption that the policy is $(\varepsilon/2, \delta)$-correct. Similarly, we must have $\mathbb{E}[T_\ell] > t^*$, for all arms $\ell > 0$. Therefore, if we have an $(\varepsilon, \delta)$-correct policy, we must have $\mathbb{E}[T] > (n/(4c\varepsilon^2)) \log(N/8\delta)$, which is of the desired form.

∎

Now we are ready to prove theorem 5.

***Proof of theorem 5.*** We consider an MDP $M$ where the transition is defined as $P_h(s'|s, a) = 1/S$ for all $(h, s, a, s')$ and **is known** to the learner. Since the action has no control over the next-state, this is equivalent to a collection of $SH$ multi-armed bandits. Due to the uniform transition, $P_h^\pi(s) = 1/S$ for any $\pi, s, h$, and so finding the $\varepsilon$-optimal policy amounts to finding an $\varepsilon_{s,h}$-optimal policy for each bandit $(s, h)$, such that $\sum_{s,h} \varepsilon_{s,h} = S\varepsilon$. Theorem 10 implies that it takes at least $\Omega(A \log(N/p)/\varepsilon_{s,h}^2)$ visits to a bandit $s, h$ to find an $\varepsilon_{s,h}$-optimal action simultaneously for each of $N$ reward functions with probability at least $1 - p$. It follows that the total number of samples required $\Omega(\sum_{s,h} A \log(N/p)/\varepsilon_{s,h}^2)$ is minimized when $\varepsilon_{s,h} = \varepsilon/H$ for all $(s, h)$, which gives a total of at least $\Omega(H^3 SA \log(N/p)/\varepsilon^2)$ samples, which translates to at least $\Omega(H^2 SA \log(N/p)/\varepsilon^2)$ episodes. ∎

## D   Proof of $N$-independent upper bound of UCBZERO in the Reward-free Setting

***Proof of Theorem 6.*** Fixing the transition kernel, we consider dividing all possible MDPs into a set of equivalence classes based on different reward patterns. Specifically, given any $M \in \mathbb{Z}^+$, we split the support of reward $[0, 1]$ into $M$ segments, $I_i = [\frac{i-1}{M}, \frac{i}{M}], \forall 1 \le i \le M$. For any MDP,

the reward function $r_h(s, a)$ depends only on state $s$ and action $a$, and for each $(s, a)$ pair, the corresponding reward must lie in one of the $M$ segments, thus there are $M^{|S| \times |A|}$ different patterns of reward functions for each step $h$, characterized by a matrix $\Phi_h \in [M]^{|S| \times |A|}$, where each entry $\Phi_h(i, j) \in [M]$ is the segment that $r_h(i, j)$ lies in. Given that we have $H$ steps, in total we will have $M^{|S| \times |A| \times H}$ different reward patterns, denoted as $\Phi = \prod_{h=1}^{H} \Phi_h$. For each $\Phi$, we next show that learning any single reward function $r \in \Phi$ is enough to cover all other reward functions in $\Phi$. Specifically, assume we have learned a near-optimal policy $\pi_r$ that satisfies

$$V_r^*(s_1) - V_r^{\pi_r}(s_1) < \varepsilon, \tag{86}$$

where subscript $r$ means the value function under reward function $r$ and $V_r^{\pi_r}$ is the value function of the learned policy. Then for any other $r' \in \Phi$ different from $r$, we have

$$V_{r'}^* - V_{r'}^{\pi_r} = V_{r'}^* - V_r^* + V_r^* - V_r^{\pi_r} + V_r^{\pi_r} - V_{r'}^{\pi_r}. \tag{87}$$

Note that

$$\begin{aligned}
V_{r'}^* - V_r^* &= \max_\pi \mathbf{E}_\pi \left[ \sum_{h=1}^H r_h'(s_h, a_h) \right] - \max_\pi \mathbf{E}_\pi \left[ \sum_{h=1}^H r_h(s_h, a_h) \right] \\
&\leq \max_\pi \mathbf{E}_\pi \left[ \sum_{h=1}^H r_h'(s_h, a_h) - r_h(s_h, a_h) \right] \leq \frac{H}{M},
\end{aligned} \tag{88}$$

where the last inequality is due to $r_h'$ and $r_h$ lie in the same segment for all $h$. Same result holds for $V_r^{\pi_r} - V_{r'}^{\pi_r}$. Let $M = \frac{H}{\varepsilon}$. Then plug (88) back to (87), and also remember that $V_r^*(s_1) - V_r^{\pi_r}(s_1) < \varepsilon$, thus we have

$$V_{r'}^* - V_{r'}^{\pi_r} < \frac{H}{M} + \varepsilon + \frac{H}{M} = \frac{2H}{M} + \varepsilon = 3\varepsilon, \tag{89}$$

which shows that the policy learned on reward function $r$ is also near-optimal for other reward functions in the same equivalence class. Given that, it suffices for our UCBZero to successfully learn a total of $M^{|S| \times |A| \times H}$ reward functions in order to cover all possible MDPs. Then simply applying the conclusion in Theorem 1 concludes the proof. ■