[Reviews · NeurIPS 2020]

Review 1

Summary and Contributions: The authors present a theoretical work that investigates sample efficient exploration in multi-task RL settings. As the paper is from the theoretical RL perspective, it deals with discrete state / action spaces in a finite time horizon (non discounted) MDP using a finite set of rewards. They propose a framework for task agnostic exploration for multi task settings in two steps: First there is an exploration phase without any reward, afterwards we want to learn to solve N tasks for which we can sample the reward for the previously collected transitions. The authors develop an algorithm: UCBZero and show that this will be able to explore near optimal in this setting (given a certain definition of optimality). At the end of the paper the authors compare their results with a recent publication in the field and are able to transfer the results to get a tighter upper bound for a similar framework.

Strengths: The paper under review provides a theoretical foundation for empirical studies of multi-task reinforcement learning, unsupervised reinforcement learning and (near optimal) exploration. Therefore it is highly relevant for this conference and the broader field. Even if the results and the derived algorithm will not be directly relevant for typical multi-task applications, like robotics, they could serve as an important foundation for future research and algorithm design. The task-agnostic framework - which they propose - is an interesting concept and is unified with the task-free (proposed by another very recent work) and task-specific view / framework by the authors at the end of the publication. They also develop bounds for their framework and show that these bounds can also contribute to the other views.

Weaknesses: As a typical publication is in the pure theoretical RL field the empirical results - even on toy tasks - are not part of the work. As this is rather common in the field it is acceptable, while still as somebody from a more practical perspective I would love to see at least a few results. In the same line, although the authors emphasise that their work is directly relevant for recommendation systems, it deals with mainly theoretical properties of MDPs that are different from what we see in typical applications (e.g. robotics). Discussing ideas how this could be made more general (continuous spaces, infinite horizon, etc. ) would have been great.

Correctness: From a high level perspective all the claims and proofs look technical sound. Empirical evaluation is not part of the paper.

Clarity: Most of the paper is generally well written and good to understand. The only issue that I see is the structure of the first chapters. Things that are introduced later are used already in the abstract and introduction, which makes them hard to understand when reading it the first time (see comments below).

Relation to Prior Work: As far as I can judge all major publications were referenced. I certainly enjoyed the detailed discussion in chapter 6 and the unification view that gives additional insight.

Reproducibility: Yes

Additional Feedback: After all, I really enjoyed reading the paper as it gives a good insight and potentially can serve as a good foundation for further research on multi-task RL in robotics. To be honest, when reading the abstract, chapter 1 and 2, the paper is really hard to understand for anybody not familiar with the topic. You are using definitions, variables etc. before you prepare the reader for what will come. E.g. the equations for the bounds in the abstract are ridiculous for anybody not already aware what the paper is dealing with. Please restructure the paper by having some preliminaries of chapter 3 before using the variables etc. After reading chapter 3 everything was much easier to understand.


Review 2

Summary and Contributions: The authors propose an algorithm for visiting all the state-action pairs in an MDP. This is performed by following an upper confidence reward scheme where the reward is set to be 0. Then, the authors show that the data that was collected from that process can be used to approximate the optimal policy for N different reward signals.

Strengths: A new algorithm for exploration with no reward.

Weaknesses: The papers fails to cite existing work on the topic, including: "Provably Efficient Maximum Entropy Exploration" Hazan et. al. 2019 "Optimistic Policy Optimization with Bandit Feedback" Shani et. al. 2020 "On-line Markov Decision Processes" Even Dar et. al. 2009. This is important since it is hard to assess the novelty of this work, and in particular, the novelty of the exploration stage. The proof for the exploration stage seems to follow a proof by Jin et. al. 2018, so it does not seem in particular novel to me. The references above are other examples for algorithms that visit all the state-action pairs enough times. Alternatively, what would have happen if we would have used a standard exploration algorithm, like UCRL2 and run it with a reward 0? would that be enough? Please clarify these issues. The second part of the paper is less clear to me, i.e., the logarithmic dependence in the number of goals. Concretely, given that we have collected enough samples from each state-action pair, it is not clear why should the bound depend on the number of goals. There also seems to be an implicit connection between the number of possible goals and the size of the MDP. Finally, why are the bounds have a ^5 (power of 5) dependence on the horizon while standard algorithms have a quadratic dependence? this should have been discussed.

Correctness: The proof of theorem 1 seems correct to me. I am less confident about the lower bound and would like the authors to clarify my concerns above.

Clarity: In addition, I did not find the explanations on the dependence in log(N) to be clear.

Relation to Prior Work: The connection to related work is lacking. I've mentioned a few papers above but there are more that are relevant.

Reproducibility: Yes

Additional Feedback: POST-REBUTTAL UPDATE: -------------------------------- I found it hard to understand what is novel in this paper. There are quite a few existing exploration solutions to visit all the states often. But these works were not compared or discussed. Concretely I gave 3 examples. While reading the authors rebuttal I understand why two of them are less relevant to their specific setup. There are, however, many more works which I did not provide in my review and are still relevant. For example, this recent paper "A Provably Efficient Sample Collection Strategy" cites many of them. This is a big problem. From reading the paper, I couldn't understand why one needs the UCBZero algorithm and can't use one of the other algorithms to visit all the state-action pairs. What does using the new proposed algorithm gives me? There were many papers in the late 90's analysing how many samples one needs from each state-action pair to get q correct estimation of the value function. In particular, Michael J. Kearns, Yishay Mansour, Andrew Y. Ng had a few papers on that: "A Sparse Sampling Algorithm for Near-Optimal Planning in Large Markov Decision Processes" "Approximate Planning in Large POMDPs via Reusable Trajectories" The last part regarding the N goals was less clear to me during the review, and the authors clarified that. This is a nice result. I personally feel that the paper would benefit from another round of reviewing and clarifications. That said, given that the other reviewers liked the paper, and that I also see many nice contributions in it, I am also fine with accepting it. The authors did a good job in addressing our comments during the rebuttal, and if they will address these concerns in the last version that might be enough. I would therefore raise my score from 5 to 6.


Review 3

Summary and Contributions: This work studies a theoretical bound of PAC-MDP algorithms in a “task-agnostic” RL setting, in which task-agnostic reward-free exploration steps are performed and then policies for N tasks are learned. The high-level idea of the algorithm is to learn a policy/value function for visiting as diverse states as possible in the spirit of UCB during an exploration phase; with a proper UCB bonus term, the PAC upper-bound can be given as O(log N H^5 SA / eps^2).

Strengths: I think the problem setting is quite interesting and the contribution has high significance in the theoretical context: practitioners would agree that task-agnostic learning is important but difficult without much theoretical guarantee so far. The setting of task-agnostic RL is closer to the practical problems than reward-free setting, as in practice instantiation of reward is only available rather than a fully given reward function, where the novelty of the theoretical framework lies in. This paper provides a theoretical foundation, many recent practical applications and algorithms involving deep RL could benefit from the framework and theory. The complexity result looks good. The log N bounds for N “dependent” tasks and linearity to S sound like a novel result. This is a clear improvement over the O(S^2A H^5 / eps^2) algorithm of [Jin et al., 2020] which is quadratic to S. The resulting algorithm is yet quite simple (UCB-style exploration), but showing this bound under a proper choice of UCB bonus would be a meaningful contribution as a basis for future work. I enjoyed the discussion about an unified view of task-specific RL v.s. Reward-free v.s. Task-agnostic RL. This would be a valuable overview for the community, clearly summarizing the current art and open questions of theoretical results on each problem setting.

Weaknesses: From a practical viewpoint the algorithm itself is not very new, which is however fine in light of the theoretical work. The task-agnostic RL framework (explore and then adopt) itself isn’t a very new idea, as already have been studied in some non-theoretical, application RL literature (unsupervised pre-training, e.g. Go-Explore [Ecoffet et al. 2019], Plan2Explore [Sekar et al., 2020], VISR [Hansen et al. 2020] to name a few). This could be briefly discussed as well. There is still a gap on H^5 bound, which is the limitation of the theoretical bound but this should be considered more as an open question. Also, the O(log N) term in the N-dependent setting would be another downside (as opposed to RFE [Jin et al., 2020]). For a practical (small) number of N, UCBZERO’s O(log N H^5) can be still higher than O(N H^2) in task-specific RL, so the benefit compared to task-specific learning can be still limited.

Correctness: Theoretical Soundness: I didn't check every step of the proof and theoretical claim, but I feel the high-level sketch of the proof is correct and the derivations are solid as far as I could check. Notation issues on Algorithm Table 1: L6 of Exploration and L5 of Policy-Optimization) Where is `t = ++N(s, a)` being used afterwards? b_t and \alpha_t should not be a parameter.

Clarity: 1. Overall, the paper looks well-written although theoretical results are not very straightforward to follow. The introduction and the motivation of the problem is presented very clearly, although the work is pretty theory-heavy. I also liked how the summary of the key theoretical results is presented in Table 1. The structure of theoretical results (background and the main algorithm) is good and easy to follow, with a proper use of appendix and summarization of the proof idea. 2. One concern is that the meaning of “N dependent” task v.s. “N independent” task doesn’t seem clearly explained (e.g. in Table 1 and Section 6), which needs to be clarified for better understanding of the difference of problem settings. For example, what is exactly being shared among dependent tasks? What is the most important attribute that enables theoretical bound as opposed to with N-independent tasks? 3. The algorithm table would be quite difficult to parse during first few iterations of reading; a small suggestion is that it would be better to have more in-English, intuitive explanation about what those equations are about, e.g., pointing to L138-L143, mentioning the terminology “pseudo-Q table” explicitly in the algorithm table itself, or saying like perform Q-learning with the UCB bonus with `b_t` (plus pointer to Thm 1). 4. Minor: It would be good to have a brief background section for reviewing how UCB-H works for the sake of self-containedness, in light of the mathematical connection between UCB-ZERO and UCB-H (L155).

Relation to Prior Work: The discussion of related work looks comprehensive. A clear summarization with an unified view reward-free vs task-agnostic framework is provided. A comparison to [Jin et al., 2020] is very detailed (related work and in section 6), which was pretty helpful. A fundamental difference is that in task-agnostic settings (this work) the true reward function is unavailable other than the instantiation upon transitions.

Reproducibility: No

Additional Feedback: Additional Suggestions: - Typo) L104 -> setting. - Please provide citations on Table 1 (for ORLC, RFE, etc.) - L47-48: Citation needed for reward bonus types. POST-REBUTTAL UPDATE: -------------------------------- My confusions and concerns on the terminology and notations were addressed in the author response, but I would suggest the authors improve the clarity and notations (will update the review accordingly). Another weakness I mentioned is the H^5 dependency (same as R3), which cannot be actually addressed in the rebuttal; I feel this is rather a scope of future work (as authors claimed), therefore it should not be a primary reason for rejecting the paper. However, I would not buy the argument “log N << S” as much, because this would be valid only if the gap between upper bound and lower bound are bridged which still remains a hypothesis. That being said, I don’t think this should be a strong reason for rejection. R3 has raised some concerns and raised good points. I mostly agree with R3 that the novelty and contribution were a bit overclaimed. However, given the main contribution of this paper lies in the theory (especially the proofs and new bound results) which I think is enough, I feel like it would be still fine to recommend acceptance, **assuming that** the authors could tone it down a little bit in the camera ready. For example, I also wouldn’t call the task-agnostic RL framework and the algorithm novel as stated in the contribution and in the title. These points would make me hesitate to give a score of strong acceptance.


Review 4

Summary and Contributions: This paper investigates exploration without the use of a reward function to guide exploration. To this end, a task-agnostic RL framework is proposed which consists of two phases: an exploration phase --- in which the agent collects a dataset of trajectories --- followed by a policy optimization phase in which the collected trajectories are augmented with the reward functions of N different tasks. An algorithm is proposed which finds epsilon-optimal policies for each of the N tasks, and it is shown that near-optimal sample complexity is achieved. The problem setting is strictly harder than both the typical task-specific and the recent reward-free exploration problems, and the theoretical results thus hold for these problem settings as well.

Strengths: My decision was primarily influenced by the following: 1) the theoretical results are novel and general, 2) the analysis suggests deeper insights which would benefit both the exploration theory community and the wider reinforcement learning community, and 3) the paper is well-written and precise. 1) As mentioned above, the problem setting generalizes both the typical task-specific exploration problem, as well as a more recent line of work on reward-free exploration. The problem setting investigated here is strictly more difficult than both of these, and their analyses apply to these prior works as special cases. 2) The analyses suggest deeper insights into the nature of exploration. In particular, theorem 1 suggests that pure exploration can be as effective as an algorithm which balances exploration and exploitation, implying that exploitation acts only as a mechanism for regret minimization rather than contributing to learning efficiency. Furthermore, if one is primarily interested in efficient multi-task learning, the analyses suggest that the simple pure-exploration approach can suffice. Finally, I suspect the results could be of interest to the planning community; the proposed algorithm explores according to a very simple optimistic initialization + UCB estimation scheme, yet the data generated is sufficient for effective model-free planning in the policy optimization phase. 3) The paper was clearly written and precise throughout. The content was clear at both a high level and a fine level of detail. Though I did not check the analyses in-depth, I did read through them and many of the proof strategies were fairly straight forward, and the lower-bound follows standard arguments from the bandits literature. I am reasonably confident in the correctness of the analyses.

Weaknesses: If I had to choose something, I guess it would potentially be difficult to apply the algorithm in practice, as it outputs non-stationary stochastic policies. That being said, I don't see this as a significant weakness of the paper given that the focus of the paper is on developing a *rigorous theoretical understanding* of the difficulties of task-agnostic RL, rather than applications.

Correctness: To the best of my knowledge the claims are correct.

Clarity: yes (see above).

Relation to Prior Work: yes, the relation to previous works is discussed in detail, and reinforced multiple times throughout the paper.

Reproducibility: Yes

Additional Feedback: I usually like to conclude with a list of minor details which could improve the quality of the writing, but honestly nothing really stuck out to me in this paper.

[Author Response · NeurIPS 2020]

We thank the reviewers for the positive feedback and their interest in our work! Below we address some questions.

**Response to Reviewer 2: Empirical evaluation**: Interestingly, we actually did an empirical evaluation in the earlier
phase of the project. We found that UCBZero can actually learn the navigation policies to all states in a 20 x 20 random
2D maze with the same number of samples required for UCB-H to learn the navigation policy to a particular target
state. Both algorithms are well-tuned for hyperparameters. We didn't include it in the submission because after all the
algorithm is only for tabular RL and is not of much empirical value to real complex problems. But we indeed hope that
our work provides some theoretical justification to the multi-task RL problem showing that efficient multi-task learning
is at least possible in the tabular setting. **Variables used before defined**: Thank you very much for pointing that out.
We will make sure to define them earlier in the paper in the revision.

**Response to Reviewer 3:** Thanks for the many technical questions! We are happy to clarify them. **Additional related
works**: [Hazan et. al. 2019] is an interesting work that we are not aware of, which explicitly aims at visiting each
state as uniformly as possible. In that sense, it is more similar to the reward-free paper of [Jin et. al. 2020], which
explicitly aims at visiting each state enough time as a subgoal of the exploration phase. Our result, however, shows
that the approach taken by Jin et. al. is too pessimistic (suffering from an additional factor of S) if the eventual goal
is still to perform tasks. The other two papers [Even Dar et. al. 2009, Shani et. al. 2020] studies the problem of
regret minimization under adversarial rewards, which is very different from our setting, where the rewards are still
stochastic but only unseen during exploration, and the goal is still to perform best policy identification, rather than
regret minimization because the regret is not even defined in our setting without a reward function. Overall, we think
it's not very precise to define UCBZero as an algorithm that merely tries to visit all the state-action pairs in an MDP.
Arguably, all provably efficient RL algorithms must visit all essential states enough times in order to guarantee either
small regret or small sample complexity. The goal of UCBZero is to perform multi-task learning even when the reward
is absent during exploration. We did show, however, that as a bi-product, UCBZero visits each state enough times
to allow other forms of downstream learning tasks. **UCRL2 with reward zero and nonoptimal dependency on H**:
These are great questions! And they are connected. Since UCBZero is essentially a zero-reward version of UCB-H,
our dependence on H essentially inherits from UCB-H, which is known to be suboptimal because of its model-free
nature. At a higher-level, our result is an example that the zero-reward version of a standard provably efficient RL
algorithm can be adapted to the task-agnostic RL setting and achieve good sample complexity. It is therefore very
reasonable to conjecture that a more efficient RL algorithm, e.g. UCRL2, if adapted in a similar way, can achieve a
better dependency on H. This is a great direction for future work. **Necessity of** $\log N$: Again, great question! The key
observation is that we assume the rewards are stochastic, so even if you have lots of data on each state-action pair, given
infinitely many stochastic reward functions, there will be one reward function whose instantiation deviates from its
mean, with a large probability. As a very simple sample, consider that you want to estimate the mean of $N$ Bernoulli
random variables simultaneously. The number of times you need to sample each Bernoulli random variable will scale
as $\log N$ due to union bound. In fact, this result is profound. It shows that the reward-free RL task defined in [Jin et. al.
2020] is impossible when the reward functions are stochastic. The bound must scale with $\log N$. All the questions are
very good, and we could see why readers may have these confusions. We will make sure to discuss all of the above
points in the revision. Thanks again for the very valuable feedback!

**Response to Reviewer 4: Empirical works in task-agnostic RL**: Yes, this is actually one of the big motivations
behind this paper. We've seen a lot of empirical works on this topic (e.g. an ICLR workshop last year), but very little
theoretical discussion, and our work aims to close the gap. We have one paragraph in the related work sections dedicated
to the empirical work and mentioned algorithms such as Go-Explore and Hindsight Experience Replay (HER), but we
are happy to add the missing ones you mentioned to the paragraph. **Dependency on $H$ and $N$**: The dependency on $H$
can potentially be improved upto $H^2$, matching the lower bound. See also response 2 to reviewer 3. The dependency
on $N$, however, is shown to be unavoidable by our lower-bound. This is due to our more realistic assumption that
only instantiations of reward are available rather than full reward function assumed in [Jin et. al. 2020]. In most
empirical problems, however, $N$ is small. For example, if one wants to learn to navigate to all states, $N = S$, and
$\log S$ is negligible compared to the other $S$ term. **Notation in Alg 1**: Sorry for the confusion. So both $b_t$ and $\alpha_t$ means
to be functions of $t$. $t = + + N(s, a)$ is used immediately in the next lines as inputs to $b_t$ and $\alpha_t$. We will make it
clearer and add the suggested comment lines in the revision! $N$ **dependent v.s.** $N$ **independent**: Sorry again for the
confusion. All tasks are always independent of each other. Here "dependent" means whether the sample complexity
depends on the number of tasks $N$, and "$N$ independent" means that the bound doesn't scale with $N$ and therefore still
holds finite even when the number of tasks goes to infinity. Thank you for pointing out the many typos and for the
polishing suggestions! We will make sure to follow them in the revision.

**Response to Reviewer 5:** Thank you for the very positive feedback on our work! We agree that the algorithm is mainly
of theoretical interest and the current gap between the empirical success of deep RL and the theoretical understanding
is still large. We hope that our paper can provide some intuition to the (fast-growing) empirical task-agnostic RL
community!

[Meta-Review · NeurIPS 2020]

This is a good paper, that requires some minor tweaks to be camera ready. All 4 reviewers supported acceptance, two knowledgeable reviewers strongly supported acceptance. R3 was the strongest critic but agreed the author response addressed their major concerns. This is a good theory paper that is well written, precise & accurate, the results provide new insights and generalize to other problem settings. The reviewers had some concern over the framing of the contributions, in particular the novelty and utility of the proposed algorithm. Please carefully review the suggestions and edit the tone of the paper.